# Basement Membranes, Brittlestar Tendons, and Their Mechanical Adaptability

**DOI:** 10.3390/biology13060375

**Published:** 2024-05-24

**Authors:** Iain C. Wilkie

**Affiliations:** School of Biodiversity, One Health and Veterinary Medicine, University of Glasgow, Glasgow G12 8QQ, UK; iain.wilkie@glasgow.ac.uk

**Keywords:** autotomy, cancer, collagen IV, Echinodermata, juxtaligamental cell, mechanical properties, metalloproteinase, metastasis, mutable collagenous tissue, Ophiuroidea

## Abstract

**Simple Summary:**

Basement membranes (BMs) are thin sheets of connective tissue that form boundary layers in the organs of most animal bodies. There is increasing interest in how changes in the strength and stiffness of BMs contribute to the normal processes by which fertilized eggs become mature adults and the abnormal processes associated with disease, such as the spread of cancerous tumors. The spread of cancer can be helped by the weakening and disruption of BMs, which is thought to result mainly from BM components being broken down by specific enzymes, although other factors may be involved. Brittlestars are marine invertebrate animals related to starfish and sea urchins. The BMs of their muscles act as tendons that link the muscles to the skeleton. Brittlestars are able to jettison their arms when they are attacked by predators. Such arm loss depends partly on the ability of muscles to detach from the skeleton due to the sudden weakening of their tendons. This contribution provides an overview of current knowledge of the structure and mechanical behavior of BMs in non-echinoderm animals and of brittlestar tendons and discusses the possible usefulness of brittlestar tendons as a model for understanding mechanisms of BM weakening in normal and disease-related processes.

**Abstract:**

Basement membranes (BMs) are thin layers of extracellular matrix that separate epithelia, endothelia, muscle cells, and nerve cells from adjacent interstitial connective tissue. BMs are ubiquitous in almost all multicellular animals, and their composition is highly conserved across the Metazoa. There is increasing interest in the mechanical functioning of BMs, including the involvement of altered BM stiffness in development and pathology, particularly cancer metastasis, which can be facilitated by BM destabilization. Such BM weakening has been assumed to occur primarily through enzymatic degradation by matrix metalloproteinases. However, emerging evidence indicates that non-enzymatic mechanisms may also contribute. In brittlestars (Echinodermata, Ophiuroidea), the tendons linking the musculature to the endoskeleton consist of extensions of muscle cell BMs. During the process of brittlestar autotomy, in which arms are detached for the purpose of self-defense, muscles break away from the endoskeleton as a consequence of the rapid destabilization and rupture of their BM-derived tendons. This contribution provides a broad overview of current knowledge of the structural organization and biomechanics of non-echinoderm BMs, compares this with the equivalent information on brittlestar tendons, and discusses the possible relationship between the weakening phenomena exhibited by BMs and brittlestar tendons, and the potential translational value of the latter as a model system of BM destabilization.

## 1. Introduction

Collagenous tissue is the dominant soft structural material in echinoderms, as is the case in most other multicellular animals. Many echinoderm collagenous structures undergo rapid, nervously mediated changes in mechanical properties, which may be reversible, e.g., the stiffening and destiffening of sea urchin spine ligaments, or irreversible, e.g., the profound weakening of syzygial ligaments in featherstar arms during autotomy (defensive self-detachment) [1,2,3]. Apart from one exception, all investigated mutable collagenous structures are assemblages of cross-banded collagen fibrils [1]. The exception is provided by brittlestars (class Ophiuroidea), in which the tendons that link the musculature to the endoskeleton are an extension of the basement membrane of the muscle cells and consist of non-fibrillar collagen. The autotomy of brittlestar arms involves the sudden destabilization and rupture of such tendons at one end of the intervertebral arm muscles at the breakage plane [4].

Brittlestar tendons can be regarded as atypical basement membranes [5]. However, basement membranes (BMs) usually take the form of thin layers of extracellular matrix that separate epithelia, endothelia, muscle cells, and nerve cells from adjacent interstitial connective tissue. BMs are ubiquitous in almost all multicellular animals, and some aspects of their composition and structure are highly conserved across the Metazoa [6,7]. There is increasing interest in the biomechanics of BMs, including the involvement of altered BM stiffness in development, aging, and pathology, particularly cancer metastasis, which can be facilitated by BM destabilization [8,9,10]. While such BM weakening has been assumed to occur primarily through enzymatic degradation by matrix metalloproteinases, emerging evidence indicates that non-enzymatic mechanisms may also contribute [7,11].

This contribution provides a broad overview of current knowledge of the structural organization and biomechanics of non-echinoderm BMs and compares this with the more limited information that is available on the tendons of brittlestar intervertebral muscles (IMTs). It focuses on the ability of BMs and IMTs to undergo changes in mechanical properties, particularly as expressed in weakening processes involving a reduction in tensile stiffness and strength. Also discussed is the possible relationship between the weakening phenomena exhibited by BMs and IMTs and the potential translational value of the latter as a model system of BM destabilization.

## 2. Supramolecular Organization

### 2.1. Non-Echinoderm BMs

In the following account, current knowledge of BM and IMT organization is summarized following the hierarchical sequence of Turro’s “supramolecular paradigm” [12], which is outlined in Table 1.

#### 2.1.1. Composition and Constitution

The composition of vertebrate BMs and the few fully investigated BMs of bilaterian invertebrates (i.e., those of the fly *Drosophila melanogaster* and the nematode worm *Caenorhabditis elegans*) is characterized by the invariable presence of collagen IV, laminin, nidogen and perlecan, a molecular assemblage that is often referred to as “the basement membrane toolkit” [5,13,14,15]. In addition, collagens XV and XVIII, or their orthologs, have been identified in all these BMs, and agrin is present in all except that of *D. melanogaster* [12,16,17,18,19,20]. Two non-bilaterian invertebrates—a placozoan and a cnidarian—have also been shown to have the full BM toolkit (augmented by collagen XV/XVIII in the latter) [6].

Collagen IV is found exclusively in BMs and is categorized as a non-fibrillar network/mesh/sheet-forming collagen. Mammals have six genetically distinct collagen IV chains (α1(IV)–α6(IV)) encoded by *COL4A1*–*COL4A6* genes. The α chain is ca. 400 nm long and comprises three domains: a short N-terminal non-collagenous 7S domain, a long middle collagenous domain consisting of Gly-X-Y tripeptide repeats that can form triple helices, and a C-terminal globular non-collagenous NC1 domain. The central domain is interrupted by 21–26 short non-collagenous regions that impart flexibility on the molecule and on the BM [21,22,23].

Collagen IV α chains form three heterotrimeric molecular isoforms (protomers): α1(IV)_2_α2(IV), α3(IV)α4(IV)α5(IV), and α5(IV)_2_α6(IV). During BM formation, two protomers associate at their C-terminal trimers to produce a dimer stabilized by covalent sulfilimine crosslinks. Four protomers then join at their N-terminal 7S regions to form a tetramer stabilized by inter-protomer covalent disulfide bonds and lysine-hydroxylysine crosslinks (Figure 1 and Figure 2A). In addition, NC1 domains associate non-covalently with triple helical domains and there are non-covalent lateral interactions between adjacent triple helices, both these types of association resulting in a network with a “chicken-wire” configuration (Figure 2B,C) [7,22,24,25,26].

Laminins are a family of glycoproteins also found exclusively in BMs. Each laminin is a heterotrimer consisting of, in mammals, one each of five α, four β, and three γ chains, there being at least fifteen different isoforms. Laminin-1 (α1β1γ1) is the most abundant isoform in mammalian BMs and undergoes calcium-dependent self-assembly to form a molecule with one long arm and three short arms (Figure 1). The long arm is an α-helical coiled coil formed from all three chains and at its distal end has cell-adhesive sites. The three short arms consist of one chain each and have globular N-terminal domains that are essential for laminin polymerization and BM assembly [22,28,29].

The earliest stage of BM formation in vertebrates and invertebrates involves the self-assembly of laminin heterotrimers into a polymeric network via non-covalent ternary interactions between the globular LN domains of one α1, one β1, and one γ1 short arm (the “three-arm interaction model”) (Figure 1). Laminin polymerization is preceded by the binding of laminins to a cell surface to allow the recruitment of other laminin-binding molecules into the developing BM. This is achieved through interactions between the LG domains of the long arms and cell membrane integrins (transmembrane receptors), α-dystroglycan (part of a complex of molecules linking extracellular components to the cytoskeleton), and sulfatides (sulfated glycolipids) (Figure 1) [14,26,30]. In the light of data from *C. elegans* and *D. melanogaster*, the role of laminins in BM assembly and organization seems to have been maintained throughout evolution [31,32].

Nidogens are glycoproteins containing three globular (G) domains separated by one link-like and one rod-like segment, thereby forming asymmetrical dumbbell-shaped molecules 30–40 nm long (Figure 1). In mammals there are two isoforms—nidogen-1 and nidogen-2—encoded by two genes. Mammalian nidogen plays a crucial role in linking the collagen IV and laminin networks through high-affinity non-covalent interactions, though there is evidence that calcium-dependent transglutaminase covalent crosslinking may also contribute. While both isoforms associate with the laminin-1 short arm chains through their G3 domains, only nidogen-1 binds to collagen IV (and perlecan: see below) through its G2 domain. The binding of nidogen to laminin becomes more important for BM stabilization during late mammalian embryogenesis and adulthood [14,33,34]. Nidogen is not essential for BM assembly in *C. elegans*, which is also the case in many, but not all, tissues of *D. melanogaster* [13,35].

Perlecan is a large heparan sulfate proteoglycan consisting of a core protein and three covalently linked glycosaminoglycan (GAG) chains. The GAG chains are located at the N-terminal domain of the core protein and interact with laminin-1 and collagen IV. The core protein consists of several modules arranged in five domains and can bind to collagen IV, the nidogen G2 domain, and the cell surface through interactions with α-dystroglycan, sulfatides, and α2β1 integrin. Mammalian perlecan is not required for BM assembly but makes a significant contribution to maintaining the structural integrity of BMs subjected to strong mechanical stress. Whereas the perlecan ortholog of *C. elegans*—UNC-52—has been detected only in the BMs of muscle cells, perlecan in *D. melanogaster* is structurally and functionally similar to that of mammals [13,14,36,37].

Like perlecan, three other BM-associated proteoglycans—agrin, collagen XV, and collagen XVIII—maintain BM homeostasis and provide collateral linkage between the laminin network, the cell surface, and the underlying cytoskeleton, in addition to having a wide range of other biological functions. Agrin has a multimodular core protein with three potential heparan sulfate attachment sites, although only two of these are thought to carry heparan sulfate chains when the protein is expressed. The N-terminus of mammalian agrin can be spliced to produce either a transmembrane form expressed in the brain or a BM-associated form containing the N-terminal-agrin (NtA) domain, which is expressed throughout the body and has a high affinity for the long arm of laminin γ1. The C-terminus contains receptor interaction sites that recognize, amongst others, integrins and α-dystroglycan. The domain organization of *C. elegans* agrin—AGR1—is very similar to that of its vertebrate orthologs [14,18,38,39].

The structurally homologous collagens XV and XVIII, also known as multiplexins, are homotrimers (α1(XV)_3_ and α1(XVIII)_3_, respectively) that have central triple helical domains with multiple interruptions of the Gly-X-Y sequence and non-collagenous C- and N- terminal domains. They are expressed ubiquitously in all vertebrate vascular and epithelial BMs and a single ortholog is widely distributed in invertebrate BMs. The multiplexins are likely to have an important structural role in maintaining BM integrity. In vertebrates, each is encoded by one gene. *C. elegans* and *D. melanogaster* each have a single multiplexin gene. The N-terminal non-collagenous domain of mouse collagen XV contains eight sites for the attachment of GAGs, which are mainly chondroitin sulfate. These sites may interact with other components of the BM extracellular matrix and with cell surface receptors. However, the structural role of collagen XV within the BM is under-investigated. The N-terminus of collagen XV (and collagen XVIII) includes a globular TSPN domain that is homologous to the N-terminal domain of thrombospondin-1, the functional significance of which is unclear. The collagen XVIII gene—*COL18A1*—generates three isoforms (short, medium, and long) differing in size, N-terminal non-collagenous region, tissue distribution, and functions. Collagen XVIII has three binding sites for the attachment of heparan sulfate chains. The C-terminal NC1 domain is common to all three isoforms and associates intermolecularly with heparan sulfate, perlecan, and the laminin-1-nidogen-1 complex. The significance of the N-terminal non-collagenous domain for the structural role of collagen XVIII is unknown [40,41,42,43,44].

Water is a significant constituent of BMs, accounting for around 90% of their volume. The hydration status of BMs is regulated mainly by the heparan sulfate sidechains of perlecan, agrin, and collagen XVIII. These sidechains consist of disaccharide units containing negatively charged carboxylate and sulfate groups that attract and retain a large quantity of water [6,42,45,46].

#### 2.1.2. Configuration and Conformation

The molecular components described above (together with others not mentioned) form a supramolecular structure whose configuration can be summarized as comprising two independent polymeric networks—one of collagen IV, which is highly crosslinked, and one of laminin, which is stabilized non-covalently. There is, however, uncertainty about the exact structural arrangement of these networks [47]. They are connected to each other mainly by nidogen and to cell surface receptors either directly (in the case of the laminin network) or via collateral linkages provided by heparan sulfate and chondroitin sulfate proteoglycans. There is evidence that BMs exhibit “sidedness”. In human adult ocular BMs, laminin is concentrated on the epithelium-facing side and collagen IV on the stromal-facing side (as depicted in Figure 1). Furthermore, in at least some BMs, the collagen IV 7S domain is orientated towards the stromal side and the NC1 domain towards the epithelial side. The asymmetrical organization of BMs results in the epithelial side being mechanically stiffer than the stromal side [47,48,49].

In histological sections of animal organs viewed in the light microscope, this supramolecular structure is usually observed as a thin sheet interposed between the basal surface of a cellular layer and adjacent interstitial connective tissue, which is difficult to distinguish unless specific histochemical methods are used, such as the periodic acid-Schiff (PAS) reaction (Figure 3A) [50]. In ultrathin sections of tissues prepared conventionally for the transmission electron microscope by glutaraldehyde fixation and heavy metal impregnation, most BMs are visualized as a ca. 30–100 nm thick electron-dense layer (lamina densa) with a finely granular and/or filamentous appearance, which is separated from the cellular plasma membrane by a ca. 30–50 nm thick electron-lucent layer (lamina lucida) bridged by fine anchoring filaments consisting of laminin [51] (Figure 3B). However, the employment of other preparative methods, such as cryofixation by slam freezing followed by freeze substitution, has shown that the lamina lucida is an artifact [52,53]. Close morphological analysis at the ultrastructural level has revealed that all investigated BMs exhibit three different structures: cords 2–5 nm thick, filaments 1.5–2 nm thick, and double tracks 4.5–5 nm thick consisting of two parallel lines separated by a lighter band [27].

### 2.2. Brittlestar IMTs

#### 2.2.1. Composition and Constitution

Like most echinoderms, brittlestars have an endoskeleton comprising interconnected plates, or ossicles, of calcium carbonate, usually in the form of a three-dimensional meshwork known as stereom. Brittlestar muscles are linked to the endoskeletal stereom by tendons that are extensions of the BM of the muscle cells. Most information on brittlestar tendons concerns those that link the intervertebral muscles to the vertebral ossicles of the arms (Figure 4A,B). These tendons (IMTs) have attracted interest because of their role in arm autotomy (defensive self-detachment), which is achieved through the drastic loss of tensile strength of (1) the ligament that connects adjacent vertebral ossicles and (2) the IMTs at one end of the intervertebral muscles (Figure 4B,C) [4,56,57]. At autotomy, the four intervertebral muscles at the plane of rupture detach from the vertebral ossicles in a pattern that is species-specific: in some species all four muscles separate from their proximal attachment sites (as in *Ophiopholis aculeata*: Figure 4); in most species the two dorsal muscles separate from their distal attachment sites and the two ventral muscles from their proximal attachment sites (as in *Ophiocomina nigra*: Figure 5). Each intervertebral muscle, therefore, has autotomy IMTs at one end and non-autotomy IMTs at the other end [58].

The only direct information on the composition of brittlestar IMTs comes from a preliminary histochemical investigation (Table 2). This shows that they contain a high proportion of carbohydrate macromolecules and that acidic groups (i.e., sulfates and carboxylates) are present, a combination that would be expected of BM-derived structures and which is attributable to the glycosaminoglycan moieties of their proteoglycan assemblage [56,57]. It was also found that treatment with *Clostridium histolyticum* type III collagenase caused the intervertebral muscles to detach from the vertebral ossicles, demonstrating both the presence of collagen in the tendons and the mechanical role of that collagen in linking the muscles to the skeleton [57].

To obtain a more detailed conception of the likely composition of brittlestar IMTs, it is necessary to make inferences based on data from conventional echinoderm BMs. Several genomic and transcriptomic investigations have confirmed that echinoderms express the full BM extracellular matrix toolkit—collagen IV, laminin, nidogen, and perlecan—as well as agrin, collagen XV, and collagen XVIII [59,60,61]. In their genomic analysis of representatives of all echinoderm classes except, unfortunately, the Ophiuroidea (brittlestars), Dolmatov and Nizhnichenko [61] found that echinoderms have two homologs of collagen IV genes, in contrast to the six of vertebrates, and five to eight laminin genes with subunits of the three types (α, β, γ) occurring amongst their gene products. They also reported that echinoderms have a single nidogen gene whose products differ from vertebrate nidogens in lacking thyroglobulin motifs (a distinction of unknown functional significance: see [62]). However, most echinoderm nidogens have an EGF-like motif at the end of their G3 (laminin-binding) domain, as is the case for the vertebrate nidogen-1 isoform [63] that binds collagen. Further similarities were observed between the domain structures of the perlecan and single multiplexin (collagens XV and XVIII) genes of the investigated echinoderms and those of their vertebrate homologs. On the other hand, most echinoderm agrins diverged in a number of ways, including lacking the NtA (N-terminal-agrin) domain that binds to laminin.

Confirmation that the collagen IV of echinoderm BMs likely forms a covalently stabilized network and, therefore, has the same mechanically critical role as that of non-echinoderm BMs was obtained by Exposito et al. [64]. Their sequencing of a collagen IV gene from the sea urchin *Strongylocentrotus purpuratus* revealed high levels of identity between its non-collagenous NC1 and 7S domains and those of human α1(IV) and α5(IV) collagen chains, these being the domains that are involved in the covalently stabilized interactions between collagen IV protomers. It was subsequently demonstrated that sulfilimine crosslinks, which provide covalent bonding at the NC1 interactions, are evolutionarily conserved in all Eumetazoa (Bilateria + Radiata), including *S. purpuratus,* as is peroxidasin—the enzyme that catalyzes sulfilimine bonding and is embedded in BMs [65].

#### 2.2.2. Configuration and Conformation

There is no direct information on the configuration (molecular disposition) of brittlestar IMTs or conventional echinoderm BMs. However, since echinoderms possess the full BM molecular toolkit, and since some of its most important components closely resemble their vertebrate homologs, particularly in the case of collagen IV and nidogen, regarding domains involved in intermolecular bonding, it is reasonable to infer that the configuration of echinoderm BMs (including brittlestar tendons) conforms to the conserved pattern observed in all investigated vertebrates and invertebrates (Table 1; Figure 1), i.e., separate collagen IV and laminin networks connected to each other and to cell surfaces by, amongst other molecules, nidogen and perlecan, the contribution of agrin being less certain.

Most information on the histology and ultrastructure of brittlestar IMTs has been obtained from the common NE Atlantic and Mediterranean species *Ophiocomina nigra*. The following account is based on references [4,57]. In histological sections stained by trichrome methods, the IMTs of *O. nigra* appear as strongly basophilic fibers that extend from the terminal regions of the muscle cells into the adjacent vertebral ossicle, where they loop round bars of skeletal stereom (Figure 6A). PAS-stained sections illustrate the continuity between the tendons and the conventional BMs of the muscle cells (Figure 6B).

No ultrastructural differences were detected between the autotomy and non-autotomy IMTs of the intervertebral muscles of *O. nigra*. Both types consist of a finely granular material within which can be discerned filaments 2–3 nm in diameter arranged roughly parallel to the longitudinal axis of the tendon fibers (Figure 7A–C). At the attachment regions, the tips of individual muscle fibers divide to form processes that penetrate into the pore spaces between the stereom bars. The tendon fibers also bifurcate and dovetail with different processes of the same muscle cell or pass between adjacent muscle fibers and continue as conventional BMs. Both IMTs and BMs are separated from the sarcolemma of the muscle cells by an electron-lucent gap up to 75 nm wide, which is bridged by fine filaments. An electron-dense cytoplasmic layer up to 80 nm thick is sometimes visible next to the sarcolemma (Figure 7A,B). The decalcified stereom bars of the vertebral ossicles are outlined by a cellular sheath. The tendon fibers loop around the stereom bars in close contact with this layer (Figure 7C). The IMTs of another species—*Ophiothrix fragilis* (which belongs to a different taxonomic order)—resemble those of *O. nigra* in terms of their ultrastructure, continuity with conventional BMs, and relationship with the skeletal stereom [66]. These features are also shared with the tendons of the brittlestar arm-spine muscles [66,67,68].

The ultrastructure of the junction between brittlestar IMTs and the adjacent muscle cell components is very similar to that of the junction between conventional BMs and myocytes in echinoderm myoepithelial tissues (see, e.g., [69,70]) and to that of vertebrate myotendinous junctions in which a BM connects skeletal muscle fibers to the fibrillar collagen of the main tendon body (see, e.g., [71,72]). In material prepared for transmission electron microscopy using conventional methods, all these junctions are characterized by a sequence of ultrastructural domains comprising an electron-dense internal lamina adjacent to the cytoplasmic side of the muscle cell sarcolemma, which, in vertebrates, consists of protein complexes connecting actin myofilaments to laminin-binding transmembrane proteins [73], the sarcolemma itself, an electron-lucent gap (the lamina lucida of BMs) bridged by fine filaments, and an electron-dense granulo-filamentous layer (the lamina densa of BMs).

Although the ultrastructure of their tendons is identical, the autotomy and non-autotomy attachment regions of *O. nigra* differ in that the former include many prominent cell processes containing large dense-core vesicles (LDCVs). These processes are located in the outer layer of the vertebral ossicle and between the muscle cells beyond the ossicle surface (Figure 8B). They resemble the juxtaligamental cell processes (JLCPs) that have been found in all echinoderm mutable collagenous structures [74]. Some of the processes have an expanded reservoir-like profile and give rise to narrow extensions containing a single row of LDCVs, which are in close contact with and parallel to muscle cells and tendon fibers. Most of their LDCVs have a circular profile 250–350 nm in diameter and medium to high electron density (Figure 8B). Such LDCV-containing processes were never found at the non-autotomy attachment regions of either the dorsal or ventral intervertebral muscles (Figure 8A). Similar JLCPs are associated with the autotomy tendons, but not the non-autotomy tendons, of the intervertebral muscles of *Amphipholis kochii* (which belongs to the same taxonomic order as *O. fragilis*) [75]. Juxtaligamental cells are specialized neurons whose perikarya are usually located in ganglion-like clusters, or nodes, that are innervated by hyponeural motor nerves. The cell bodies of the tendon-associated JLCPs of *O. nigra* are located remotely in nodes at the ventral surface of each vertebral ossicle. A thick nerve-like trunk of JLCPs ascends from each node and branches repeatedly inside the vertebral ossicle to form a network extending throughout autotomy attachment regions [74,76].

Wilkie and Emson [4] and Wilkie [57] compared the ultrastructure of intact autotomy IMTs of *O. nigra* with that of IMTs fixed during the process of autotomy and after autotomy had been completed. In sections of material fixed during autotomy, those extremities of muscle cells that are normally located in the vertebral stereom had separated from the ossicle. Although some tendons had ruptured, others had not and had consequently been stretched considerably beyond the surface of the ossicle. Also visible were single rows of membrane-bounded vesicles with sparse granular contents, which were in close contact with the tendon (Figure 8C,D). In sections of material fixed after autotomy, there was complete separation of the muscle from the vertebral ossicle. Tendon fragments were attached to the tips of detached muscle cells (Figure 9A), and at the ossicle attachment regions, tendon vestiges appeared as U-shaped loops whose ruptured ends projected beyond the ossicle surface (Figure 9B; for histological appearance, see Figure 6C). Lined up against the tendon stumps were rows of membrane-bounded vesicles with circular to oblong profiles of variable size and with sparse, finely granular contents (Figure 9B,C). In some cases, it appeared that the rows of vesicles were attached to, or emanated from, JLCPs, suggesting that they were juxtaligamental LDCVs that had either been extruded or left exposed by dispersion of the JLCP plasma membrane and that their contents had been depleted or altered in a way that reduced their electron density (Figure 9B,C).

### 2.3. Comments on Supramolecular Organization

All available evidence indicates that brittlestar IMTs are BM derivatives. As well as being continuous with the conventional BMs at the lateral surfaces of intervertebral muscle cells, their histochemistry and the ultrastructural similarities between the tendon-muscle junctional complex and the conventional BM–muscle complexes of echinoderms and vertebrates are conducive to such an interpretation.

Current knowledge of the supramolecular organization of brittlestar IMTs is very limited. However, in view of the genomic and transcriptomic information on the molecular constituents of echinoderm BMs, it is a reasonable working hypothesis that, despite their atypical conformation, IMTs share the widely conserved features of BM composition, constitution, and configuration (Table 1; Figure 1).

Similar muscle–ossicle junctions consisting of tendinous loops emerging from the BM of muscle cells and interlacing with skeletal stereom occur in starfish (Asteroidea) [66,77,78]. BMs are also components of muscle–ossicle junctions in sea urchins (Echinoidea) and sea cucumbers (Holothuroidea), although, in those classes, they do not link directly with the stereom [66].

The employment of BMs as tendons directly connecting muscles to a hard skeleton, and therefore being solely responsible for force transmission between these structures, appears to be unique to brittlestars and starfish. However, only brittlestars have mechanically adaptable tendons that can undergo an endogenous weakening process that permits rapid muscle detachment at autotomy. This phenomenon is discussed in Section 3.2.2 below.

## 3. Biomechanical Aspects

### 3.1. Non-Echinoderm BMs

#### 3.1.1. General Biomechanical Aspects

BMs have diverse physical functions. They compartmentalize tissues, act as sieves that restrict the movement of cells and molecules, and link cells to interstitial connective tissue. In addition, they perform the same general mechanical functions as other collagenous structures, i.e., they resist, transmit, and dissipate mechanical forces, and they store and release elastic strain energy. The growing realization of the importance of BM biomechanics in normal and pathological processes has led to the proliferation of studies characterizing their mechanical properties. These endeavors have been assisted by an expanding array of nanomechanical testing methods, such as atomic force microscopy, magnetic nano-tweezers, and Brillouin microscopy [79,80,81].

BM-related biomechanical principles have been explained in detail by Khalilgharibi and Mao [7], Chang and Chaudhuri [11], and Krag and Andreassen [82]. The main biomechanical parameters pertinent to the subsequent discussion are (1) tensile stress—tensile force per unit of cross-sectional area; (2) ultimate tensile strength—maximum tensile stress that a structure can withstand without breaking; (3) tensile strain—change in length expressed as a proportion of initial length; and (4) tensile stiffness (Young’s modulus)—rate of increase in stress sustained by a structure subjected to increasing strain.

Published data on the tensile stiffness of BMs, which is their most frequently reported mechanical property, show a wide range of values, although most are around or above 1 MPa (Table 3). The stiffness of cellular layers alone is likely to be around 1 kPa [83,84], from which it can be inferred that externally imposed deformation of intact cellular tissues results in most of the stress being sustained by the BM. The wide variability in the stiffness values is likely to be due in part to differences in the organization and functional requirements of BMs in different species and at different locations and in part to differences in measurement techniques or procedural details [11]. Table 3 and Table 4 also show that the values for BM stiffness (0.002–10 MPa) and ultimate tensile strength (0.17–2 MPa) lie within, and at the lower end of, the ranges reported for intact connective tissue structures composed of fibrillar (mainly type I) collagen (0.0004–1430 MPa and 0.14–95 MPa, respectively). The latter ranges are considerably lower than the estimated stiffness and tensile strength of collagen molecules (3–9 GPa and 11 GPa, respectively), an indication of the extent to which the mechanical behavior of collagenous structures is influenced by intermolecular interactions.

Since it appears that the only covalent intermolecular crosslinks contributing to the BM constitution are those that stabilize the collagen IV network, there is an a priori expectation that this network is the main determinant of BM mechanical properties, at least at shorter timescales (minutes to hours) [7]. Some investigations have corroborated this empirically. For example, the importance of the collagen IV network for BM tensile strength is demonstrated by the weakening of mammalian capillary walls and consequent hemorrhage associated with collagen IV gene mutations [111,112]. Knockdown of the *D. melanogaster ColIVα1* gene results in a strong reduction in the stiffness of the egg chamber BM, which is not observed with laminin, nidogen, or perlecan knockdown [113]. Reducing the density of covalent sulfilimine crosslinks, which connect the C-terminuses of collagen IV protomers, destabilizes *D. melanogaster* egg chamber and larval midgut BMs and decreases BM stiffness in the mouse renal tubule [89,114,115]. However, other studies have demonstrated that laminin can contribute significantly to BM stability [47]. In one of these [116] it was found that laminin deficiency destabilized a mouse retinal BM (the inner limiting membrane) even though the relative abundance of collagen IV was unchanged.

#### 3.1.2. Mechanical Adaptability

BMs are dynamic structures that, during the lifespan of an individual animal, undergo changes in organization and mechanical behavior associated both with normal aging, development, and homeostasis, and with the causation and progression of disease [7].

Evidence for age-related changes has been obtained from human ocular BMs. Krag and Andreassen [82] found that the ultimate tensile strength of the anterior lens capsule decreases from 17.5 MPa in childhood to 1.5 MPa in the elderly. On the other hand, under low strains (within the accommodative range), the stiffness of the anterior lens capsule and inner limiting membrane increases with age [45,82]. These changes may result from alterations in BM composition; with advancing age, the relative concentrations of collagen IV and agrin in the inner limiting membrane increase, and that of laminin decreases slightly [45]. It is not known if such age-dependent biomechanical changes occur in BMs at other locations. There is also apparently conflicting information on compositional changes; while it was reported that the relative collagen IV content of human cerebral microvessels increases with age [117], expression of *COL4A1* and *COL4A21* genes decreases with age in mouse cortical microvessels [118].

*D. melanogaster* provides several examples of changes in BM tensility that occur during developmental processes (reviewed in [119]). For instance, epithelial folding of developing wing imaginal discs depends partly on a reduction in BM stiffness that results from the site-specific cleavage of collagen IV at triplet interruption regions [120,121]. BM stiffening plays an important role in the morphogenesis of the *D. melanogaster* egg chamber. Over 2–3 weeks, the whole BM, which is the outermost layer of the egg chamber, shows an increase in stiffness that is much more pronounced in the central region than at the poles. The egg chamber is initially spherical, but as it grows in size, it elongates due to the greater resistance of the BM to circumferential expansion than to longitudinal expansion. The increased stiffness is associated with changes in composition and conformation whereby new fibril-like structures are incorporated into the BM [92,93]. Early stages of mouse morphogenesis also involve BM stiffening. Soon after implantation, the BM in the embryonic half of the “egg cylinder” exhibits an even distribution of perforations that renders the BM compliant enough to accommodate embryonic growth. The perforations result from the expression of the matrix metalloproteinases MMP2 and MMP14, which degrade collagen IV and laminin. Immediately before gastrulation, over the course of around a day, the perforations and MMP expression become localized to the posterior side of the embryo. This increases the stiffness of most of the embryonic BM but leaves a posterior zone of least resistance. Gastrulation starts with the progressive rupture of the BM at this region of weakness by forces imposed by cell proliferation and tissue growth that give rise to the primitive streak [9].

In vertebrates, MMP-generated defects and the consequent reduction in stiffness of the BM have a widespread role in the morphogenesis of branching organs such as lungs and salivary glands by permitting the expansion of growing bud tips [7]. A similar mechanism may facilitate the transmigration of leukocytes across blood vessel walls during immune surveillance in response to inflammatory signals; there is evidence that neutrophils employ MMP-mediated remodeling of the vascular BM to weaken it and increase its deformability [122]. Neutrophils are also mainly responsible for the abnormal MMP9 levels that lead to collagen IV degradation and weakening of the microvascular BM associated with hemorrhagic events after human ischemic stroke [123,124].

Cell invasion through epithelial or endothelial BM barriers marks the initiation of metastatic cancer [125]. Ever since it was demonstrated that cancer cells could degrade collagen IV and that increased degradation correlated with greater metastatic potential [126], the established view has been that BM invasion occurs primarily through MMP-dependent proteolysis [11]. The envisaged mechanism is that selective proteolysis, particularly of the covalently stabilized collagen IV network, weakens the BM, facilitating its plastic (i.e., partly irreversible) deformation and ultimate rupture by mechanical stress generated by cell division or general tumor expansion [11]. Many cancer cells deliver MMPs to the BM extracellular matrix by means of invadopodia—F-actin-based plasma membrane protrusions that secrete MMP2 and MMP9 by exocytosis and express MMP14 as a transmembrane protein [127,128]. Invadopodia are, however, active structures that can exhibit repeated protraction–retraction cycles over several hours. This, in turn, generates protrusive and contractile forces, which, it has been proposed, could work synergistically with proteolysis to break down the BM [11]. This idea has received support from the investigation of the invasion by anchor cells (specialized uterine cells) into the vulval epithelium of *C. elegans* during development. While MMPs promote BM invasion in vivo, this process still proceeds in MMP-negative animals due to the compensatory formation of large cellular protrusions that breach the BM and displace it through the application of compressive force (Figure 10) [125]. This highlights the need for vigilance in looking out for other protease-independent mechanisms of BM destabilization that may operate in pathological and non-pathological processes of cell transmigration, especially since these would be potential targets for therapeutic intervention.

### 3.2. Brittlestar IMTs

#### 3.2.1. General Biomechanical Aspects

The main mechanical function of tendons is to transfer contractile force from muscles to hard skeletal components. To do this efficiently, i.e., with minimal loss of energy, tendons have to stretch as little as possible under the pattern of forces (in terms of magnitude, rate of application, and duration) to which they are subjected in vivo and any lengthening that does occur has to be reversed through passive elastic recoil. However, vertebrate tendons, which consist mainly of fibrillar collagen I, are employed in two main ways: those that are thick and short compared with their muscles stretch little under load and transfer contractile force efficiently, as just described; those that are long and thin compared with their muscles stretch considerably under load and store elastic strain energy that can be released to supplement muscular force, thus providing an energy-sparing mechanism [129,130]. While brittlestar IMTs are not thick in comparison with the individual muscle fibers (see, e.g., Figure 7A), it is obvious from Figure 6A,B that their relative length is very small; for example, if IMT length is measured as the maximum depth of penetration of tendon loops into the skeletal stereom, it is at most 2% of the length of the muscle fibers. It is, therefore, likely that IMTs resemble functionally the first vertebrate type described above and that they transfer force efficiently with negligible lengthening and do not function as elastic energy stores.

Vertebrate tendons, like other collagenous structures (including BMs), are viscoelastic and show strain rate dependence. At low strain rates (when they are stretched slowly), they are more deformable, lose more energy as heat due to their constituent collagen fibrils and fibers (fibril bundles) shearing past each other, and are less effective at transferring force; at high strain rates, they are less deformable, lose less energy through interfibrillar or interfiber shear, and transfer force more effectively [7,131,132]. The intervertebral muscles of brittlestars are responsible for the rapid bending movements of the arms that are observed during, for example, locomotion, some food-capturing strategies, and the thrashing reflex of autotomized arms [57,133,134,135]. During such movements, the duration of the intervertebral muscle contraction–relaxation cycles is in the order of 1 s (see, e.g., [134,135]). The tendons of these muscles are, therefore, subjected to high strain rates and would be expected to mediate efficient force transmission. Another feature of the biomechanical environment of these tendons is that they are never exposed to prolonged tensile loading, which could result in stress relaxation or irreversible deformation (creep) [136]. Although brittlestars can maintain rigid arm postures for extended periods of time, for example, during suspension feeding or as a result of defensive “freezing” reflexes [133,137], such postural fixation depends on the reversible stiffening of their mutable intervertebral and outer arm plate ligaments rather than on sustained contraction of the intervertebral muscles [138,139].

Little can be inferred about the mechanical strength of brittlestar IMTs. The ultimate tensile strength of vertebrate tendons is always higher, and sometimes considerably higher (e.g., by a factor of 4), than the maximum tensile stress to which they are exposed in vivo [140]. As contractile stresses reported for echinoderm muscles range from 20 kPa to 60 kPa [141,142], it can be tentatively suggested that brittlestar intervertebral muscles generate tensions within the same range and, therefore, that the ultimate strength of IMTs is higher than 100 kPa, which does not conflict with the range of BM values shown in Table 4.

#### 3.2.2. Mechanical Adaptability

As was explained above (Section 2.2.2), histological and ultrastructural observations indicate that during brittlestar arm autotomy, intervertebral muscles at the fracture plane break away from the vertebral ossicles through the rupture of their autotomy tendons. Evidence that this involves a change in the mechanical properties of these tendons is based on the ultrastructural appearance of the participating attachment sites in *O. nigra* material that was chemically fixed whilst undergoing autotomy. Ultrathin sections of such material show that the tendons stretch markedly before undergoing complete rupture (Figure 8C) [4]. It can be assumed that such elongation is not the usual response of the tendons to contraction of the intervertebral muscles since this would be incompatible with effective force transfer during the actuation of normal arm movements (Section 3.2.1). In *O. nigra*, arm detachment, and therefore tendon elongation and rupture, usually occur 1 s or less after the onset of autotomy-inducing stimulation (range 0.4–5.4 s; mode 0.6 s) [57]. Tendon elongation cannot, therefore, be attributed to the muscles generating a supranormally powerful contractile stress since in a timescale of ca. 1 s and assuming tendon stiffness and strain rate dependency did not change, this would result in brittle failure preceded by minimal creep. Stretching of the tendons at autotomy must, therefore, result from an endogenous reduction in their stiffness that is sufficient to permit their deformation by intervertebral muscles applying the normal range of contractile force [4].

The molecular mechanism responsible for the destiffening of autotomy IMTs is unknown. It is tempting to assume that it is related to the mechanisms underpinning the mechanical adaptability of other echinoderm mutable collagenous structures. However, all these other structures consist of interstitial connective tissue that is dominated by fibrillar (mainly type I) collagen and has a supramolecular organization fundamentally different from that of IMTs [1,143]. Furthermore, since interstitial connective tissue and BMs have separate phylogenetic and embryogenetic origins [6,144,145], the molecular bases of their respective capacities for tensile change must have evolved independently. This does not preclude the possibilities that they share features due to convergent evolution or that they exploit common principles.

Regarding understanding of the molecular basis of variable tensility in fibrillar MCT, the current prevailing view, supported by evidence obtained almost entirely from sea cucumber dermis, is that fibrillar MCT tensility is regulated by secreted effector molecules that directly influence interactions between extracellular components at different hierarchical levels [143,146,147,148]. Potential effector molecules that have been isolated from sea cucumber dermis include the tensilins, which have a high degree of sequence identity to TIMP (tissue inhibitor of metalloproteinase) proteins and cause aggregation of isolated collagen fibrils, and softenin, which acts as a destiffener, possibly by competing for tensilin binding sites on collagen fibrils. Central to this model is the concept that the collagen fibrils of these tissues are connected by labile, non-covalent linkages that can be manipulated under physiological control. Since a mechanically critical component of brittlestar IMTs is probably a collagen IV meshwork stabilized by covalent bonds (Section 2.1.1), it would appear that the fibrillar MCT model can provide little insight into its IMT analog.

One notable feature that is shared by mutable fibrillar collagenous structures and brittlestar autotomy tendons is their close microanatomical apposition to the LDCV-containing processes of juxtaligamental cells. It is highly likely that these specialized neurons are the effector cells that directly modulate the tensile behavior of MCT. They are absent from non-autotomy brittlestar tendons and from non-mutable fibrillar structures; they are the terminal components of motor pathways from the central nervous system; and no other cell type associated with MCT shows connectivity to the nervous system or has a distribution and abundance that match the rapidity and global nature of the tensile changes undergone by MCT. In addition, there is evidence that these cells synthesize effector molecules since tensilin and another stiffening factor—stiparin—have been immunolocalized to the LDCVs of juxtaligamental cells in sea cucumber dermis [143].

As remarked above (Section 2.2.2), ultrastructural observations suggest that during autotomy, the contents of LDCVs in the JLCPs of autotomy IMTs are depleted or undergo some other change. This may signify the release of a destabilizing agent that causes IMT destiffening. Although the identity of such an agent can only be surmised, it is difficult to envisage how destiffening, elongation, and rupture of the IMTs could be achieved without prior disruption of their laminin and/or collagen IV frameworks and, therefore, without protease activity. The ultrastructural appearance of elongated tendons does not indicate that they undergo gross dissolution (Figure 8C,D), suggesting that, if proteolysis occurs, it is targeted and, thus, could involve secreted MMPs. It is intriguing that the only known example of conventional BM destabilization in echinoderms—that occurring during early morphogenesis in the sea urchin blastula—is, as in vertebrates (Section 3.1.2), MMP-dependent, the main substrate of the enzymes responsible—SpMMP14 and SpMMP16—being laminin. These enzymes are, however, transmembrane, not secreted, proteins, and BM degradation occurs over a period of 2–3 h [145,149,150,151].

Neural influences on the integrity of collagenous tissue are not unprecedented. MMPs expressed by neurons of mammalian central and peripheral nervous systems are involved in the proteolysis of extracellular components associated with the extracellular matrix remodeling that accompanies neuronal growth and nervous system development [152]. Neuronal MMPs can be expressed constitutively and stored in intracellular vesicles, including the dense-core vesicles of neurosecretory cells [153,154,155]. There is, therefore, the potential for them to be released rapidly and contribute to processes operating within the short timescales associated with MCT destabilization at autotomy.

### 3.3. Comments on Biomechanical Aspects

The wide range of values for BM stiffness shown in Table 3 is likely to be related in part to the varying patterns of force imposed on the cellular layers attached to BMs and to the deformability required of these layers in fulfilling their physiological functions. Most BMs are subjected predominantly to biaxial tension, for example, when tissue sheets are indented or hollow anatomical structures are inflated. In such circumstances, BMs determine both the rate of deformation and the maximum deformation achievable by the applied force. In contrast, brittlestar tendons are subjected only to uniaxial tension resulting mainly from contractile force acting in a direction parallel to the longitudinal axis of the tendons. The parallel, longitudinally arranged 2–3 nm filaments that are a prominent feature of IMT ultrastructure (Figure 7C and Figure 8D) may be an adaptation for stiffening IMTs along the line of action of the contractile force, thereby optimizing IMT-mediated force-transfer.

The destabilization of BMs associated with development, homeostasis, and cancer metastasis is achieved largely by MMP-dependent degradation targeted mainly at laminin and collagen IV networks. It is a reasonable working hypothesis that the mechanism of IMT destiffening and rupture at autotomy is also MMP-mediated. There is, however, a pronounced difference between the timescales of the processes to which BM and IMT destabilization contribute. Regarding the former, the shortest reported time courses are shown by cellular invasion events during early gastrulation and cancer metastasis, which transpire over a few hours, whereas IMT destiffening and rupture take ca. 1 s. This discrepancy may be due to the very different dynamics of the respective processes rather than to the destabilization mechanisms themselves. BM disruption facilitates cell transmigration events regulated primarily at the level of gene expression [150,156], whereas IMT rupture is part of a defensive behavioral response that is nervously instigated and that, to be effective, has to be completed as quickly as possible. This raises the question of whether a mechanism based on enzymatic cleavage could achieve the required effect in ca. 1 s. It seems plausible that, once activated enzymes reach IMTs, significant collagen IV degradation could be achieved in under 1 min, in view, for example, of the rate of proteolysis recorded in experiments investigating the cleavage of model collagen trimers by MMP1 [157]. However, the intracellular trafficking and secretory processes responsible for the extracellular appearance of activated enzymes are slower; it may take a few minutes for secreted and membrane-type MMPs to become available for proteolysis after the onset of stimulation [158,159]. It is, of course, possible that the ultrastructural changes exhibited during autotomy by the JLCPs of the autotomy IMTs (Figure 8C,D and Figure 9B,C) are manifestations of a unique cellular mechanism for accomplishing the fast and extensive release of a protease, or other destiffening agent, into the extracellular vicinity of the IMTs. As echinoderms are notorious (and celebrated) for “doing things differently”, researchers should always be on the lookout for unexpected strategies.

## 4. Conclusions

There is a huge disparity between our knowledge of the organization and physiology of conventional BMs on the one hand and brittlestar IMTs on the other. The explanation for this is obvious: conventional BMs are ubiquitous in the bodies of vertebrates and most other metazoans and play all-pervading roles in human health and disease, whereas IMTs are obscure microanatomical peculiarities present in a single taxonomic class comprising non-iconic and relatively under-investigated marine invertebrates. The ultimate aim of the present review is to lessen this informational imbalance by encouraging further investigation of IMTs. This is a worthwhile endeavor for two reasons. First, IMTs are of inherent scientific interest; as BM-derived structures, they have unique functionality in being entirely responsible for force transfer between muscles and skeletal components and in having the capacity to destiffen drastically and irreversibly in a timescale of ca. 1 s. Second, their investigation may provide signposts to previously unsuspected mechanisms of pathological and non-pathological BM destabilization, which could eventually lead to the development of novel therapeutic approaches. As Kelley et al. [122] commented, “…many important mechanisms underlying basement membrane transit have likely been overlooked”. IMTs, therefore, have potential translational value.

A priority for research on IMTs should be the detailed characterization of their molecular composition, perhaps employing first a genomic approach to determine if brittlestars have the full BM toolkit and if they conform to the echinoderm pattern already identified in the other four echinoderm classes [59,60,61]. It is helpful that the genome of the brittlestar *Ophioderma brevispinum* has recently become available [160]. A comparative proteomic analysis of autotomy and non-autotomy IMTs or attachment regions would provide more information on their composition and might also reveal clues to the molecular basis of the former’s mechanical adaptability. In view of their putative role as the effectors of tensile change, proteomic and transcriptomic analyses of the juxtaligamental cells are required to profile their synthetic and secretory repertoire and detect candidate effector molecules. In at least some brittlestar species, the juxtaligamental perikarya, whose processes extend to the autotomy IMTs, are surgically accessible, though other juxtaligamental nodes present less of a challenge and might be more tractable subjects for the exploration of juxtaligamental cell function [57,74].

The IMTs themselves, particularly those at the dorsal edge of the dorsal intervertebral muscles, are easily accessible in some species via minor surgery (see Figure 5) [57] and are, therefore, amenable to in vitro experimentation. The direct measurement of IMT tensile properties may be achievable by employing, for example, atomic force microscopy, which has been applied successfully in the investigation of BM mechanics [84,92,93]. If this methodology proved to be practicable, it could be the basis of a model system that allows the direct experimental manipulation of autotomy and non-autotomy IMTs. It could be used, for example, to test the targeted proteolysis hypothesis (Section 3.2.2) by determining the effects of specific protease inhibitors.

As unique BM-derived structures whose mechanical properties are neurally modulated, brittlestar IMTs represent yet another enigmatic variation on the theme of echinoderm “strangeness” [161]. In 1680, the English physician and comparative morphologist Edward Tyson wrote [162]: “In every Animal there is a world of wonders”. The present author hopes that this review will convince readers that brittlestar IMTs are one such wonder that deserves more scientific attention than it has received hitherto.

## Figures and Tables

**Figure 1 biology-13-00375-f001:**
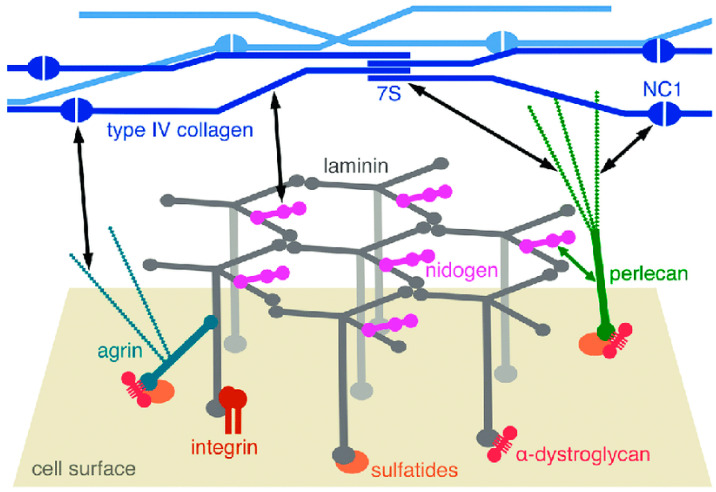
Diagrammatic representation of the molecular structure of a BM. The laminin network is anchored to the cell surface by interactions of the long arms with cellular receptors–integrins, α-dystroglycan, and sulfatides (sulfated glycolipids). There are collateral interactions with the heparan sulfate proteoglycans agrin and perlecan. Type IV collagen forms another network through interactions of its N-terminal 7S and C-terminal NC1 domains and through lateral associations of the triple helices. The laminin and collagen networks are linked by nidogen and heparan sulfates, as indicated by the black double-headed arrows. From reference [26]. Copyright © 2012 The Authors. Published by Informa UK Limited (London, UK), trading as Taylor & Francis Group.

**Figure 2 biology-13-00375-f002:**
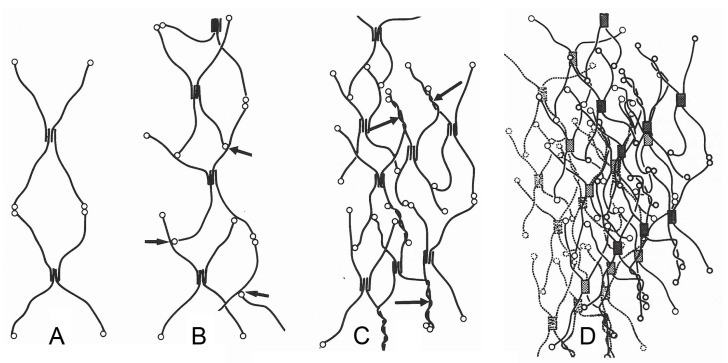
Schematic model of the type IV collagen network of a BM. (**A**) Interlinkage of type IV collagen protomers. (**B**) Binding of C-terminal NC1 domains of type IV collagen protomers at other protomer segments (arrows). (**C**) Additional supertwisting of protomers (arrows) with anastomosing and branching. (**D**) Projection of several type IV collagen networks on top of each other. 7S, N-terminal domain; NC1, C-terminal domain. From reference [27]. Used with permission of John Wiley & Sons–Books, conveyed through Copyright Clearance Center, Inc. (Hoboken, NJ, USA).

**Figure 3 biology-13-00375-f003:**
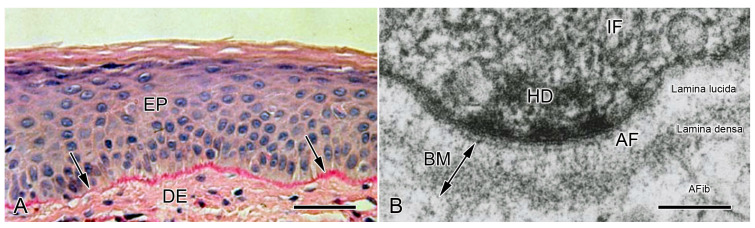
Conformation of basement membranes as observed in the light microscope (LM) and transmission electron microscope (TEM). (**A**) LM. Vertical histological section of human epidermis. Arrows indicate BM. DE, dermis; EP epidermis. Scalebar = 50 µm. From reference [54]. Used with permission of McGraw Hill LLC, conveyed through Copyright Clearance Center, Inc. (**B**) TEM. Vertical ultrathin section of human epidermis, showing the basal region of a keratinocyte (top) and its adjacent BM. AF, anchoring filaments (mainly laminin); AFib, anchoring fibrils (mainly collagen VII); HD, hemidesmosomal plaque; IF, intermediate filaments. Scalebar = 100 nm. From reference [55]. Reproduced with permission from the American Academy of Dermatology. Copyright 2023. All rights reserved.

**Figure 4 biology-13-00375-f004:**
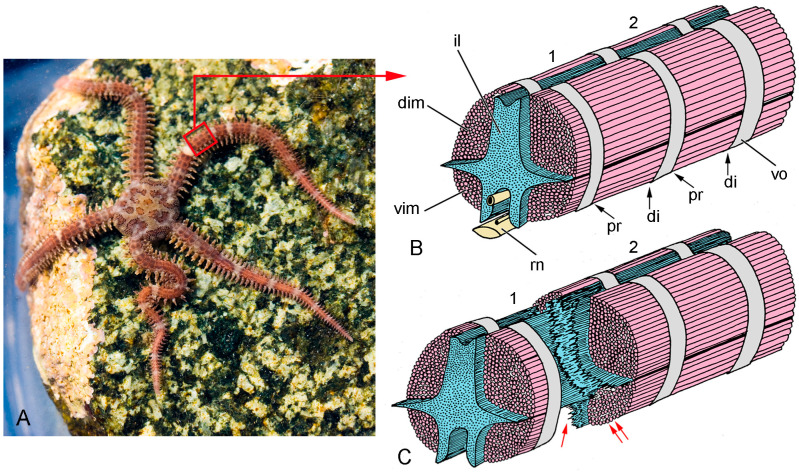
Gross anatomy and autotomy response of the brittlestar *Ophiopholis aculeata.* (**A**) Living individual, New England, USA. (**B**,**C**) Diagrammatic representations of the interior of the arm of *O. aculeata* after removal of the sheath of outer arm plates (distal end to right), showing two complete arm segments (1, 2). Adjacent vertebral ossicles (vo) are connected by an intervertebral ligament (il) and paired dorsal (dim) and ventral (vim) intervertebral muscles. The proximal (pr) and distal (di) ends of each muscle are linked to a vertebral ossicle by IMTs (not represented). (**B**) Both arm segments are intact. (**C**) Arm segment 1 is undergoing autotomy (defensive self-detachment), during which the intervertebral ligament and proximal IMTs are drastically weakened, resulting in the rupture of the ligament (single red arrow) and separation of the intervertebral muscles from the vertebral ossicle (double red arrows) respectively. rn, radial nerve cord. (**A**) Photograph by Ken-ichi Ueda, https://commons.wikimedia.org/wiki/File:Ophiopholis_aculeata_17283.jpg (accessed on 22 February 2024), under the terms of the Creative Commons License, https://creativecommons.org/licenses/by/4.0/deed.en, accessed on 22 February 2024.

**Figure 5 biology-13-00375-f005:**
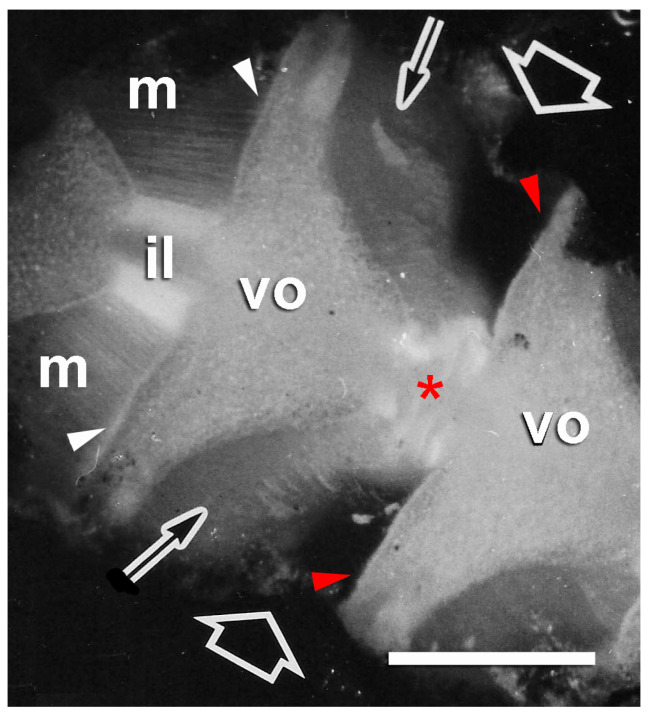
Autotomizing arm of brittlestar *Ophiocomina nigra*. Dorsal side of a short length of arm from which the dorsal arm plates have been removed to reveal the dorsal surfaces of vertebral ossicles (vo), dorsal intervertebral muscles (m), and intervertebral ligaments (il). Autotomy is occurring at the intersegmental joint indicated by the fat arrows, where the intervertebral ligament is disintegrating (asterisk) and the muscles (thin arrows) have separated cleanly from the vertebral ossicles at their distal attachments (red arrowheads). The muscles in an adjacent segment are still connected to the vertebral ossicles at their distal attachments (white arrowheads). Scalebar = 0.5 mm. Adapted from reference [57]. Used with permission of John Wiley & Sons—Books, conveyed through Copyright Clearance Center, Inc.

**Figure 6 biology-13-00375-f006:**
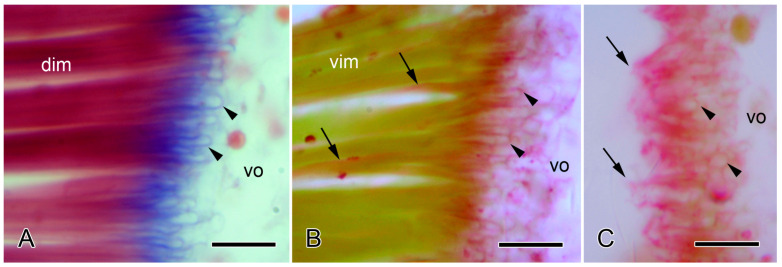
Light microscopy of the autotomy IMTs of the brittlestar *Ophiocomina nigra*. Horizontal 8 µm-thick sections of the autotomy IMTs at junctions between intervertebral muscles (dim, vim) and vertebral ossicles (vo). Scalebars = 10 µm. (**A**) Distal junction of dorsal intervertebral muscle (dim) stained with Milligan’s trichrome. IMTs are stained blue. Tendon loops (arrowheads) enclose (decalcified) bars of skeletal stereom. (**B**,**C**) Proximal junctions of ventral intervertebral muscle (vim) stained with periodic acid Schiff’s reagent and counterstained with Orange-G. (**B**) Intact junction, showing PAS-positive tendon loops (arrowheads) extending from the ends of the muscle fibers into the ossicle and PAS-positive basement membranes (arrows) at the lateral edges of the muscle fibers. (**C**) Junction after autotomy. The muscle fibers have separated from the ossicle, and only ruptured tendons (arrows) remain; arrowheads indicate tendon loops within the vertebral ossicle.

**Figure 7 biology-13-00375-f007:**
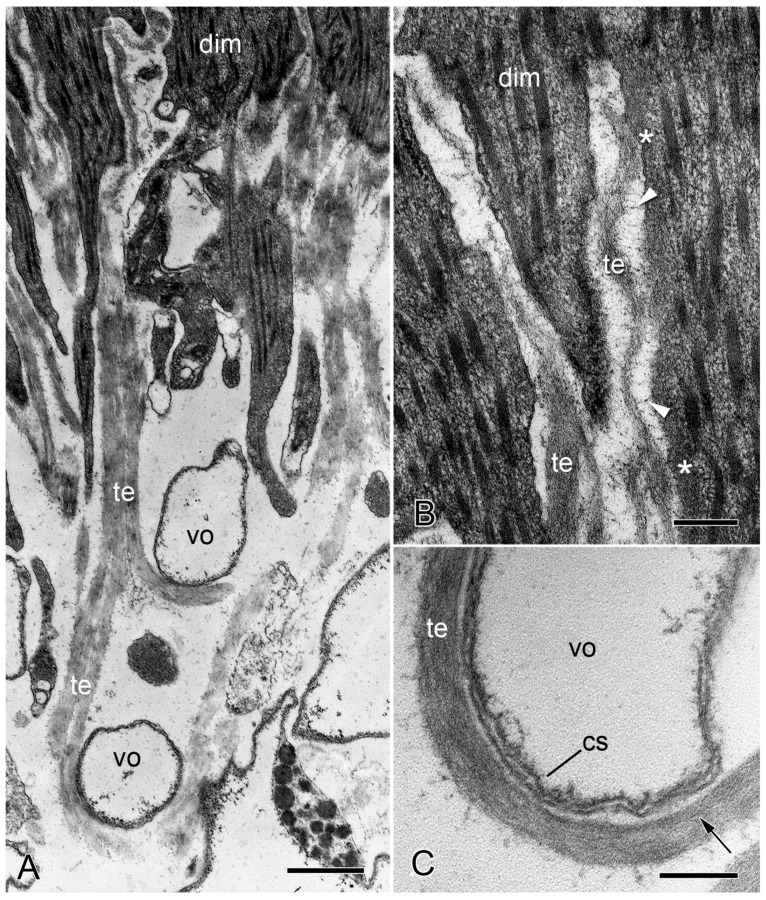
Transmission electron microscopy of the IMTs of *O. nigra*. Horizontal sections through the proximal non-autotomy attachment region dorsal intervertebral muscles (dim). (**A**) Low magnification view showing tendon fibers (te) looping around two bars of decalcified skeletal stereom (vo). Scalebar = 0.5 µm. (**B**) Attachment of IMTs to muscle cells. Asterisks indicate the electron-dense subsarcolemmal layer. The electron-lucent gap between the IMTs and muscle cell is bridged by fine filaments (arrowheads). Scalebar = 0.2 µm. (**C**) Tendon loop containing longitudinal filaments (arrow) and in close contact with the cellular sheath (cs) outlining a decalcified stereom bar. Scalebar = 0.2 µm. (**A**,**B**) From reference [4]. (**C**) From reference [57]. Used with permission of John Wiley & Sons—Books, conveyed through Copyright Clearance Center, Inc.

**Figure 8 biology-13-00375-f008:**
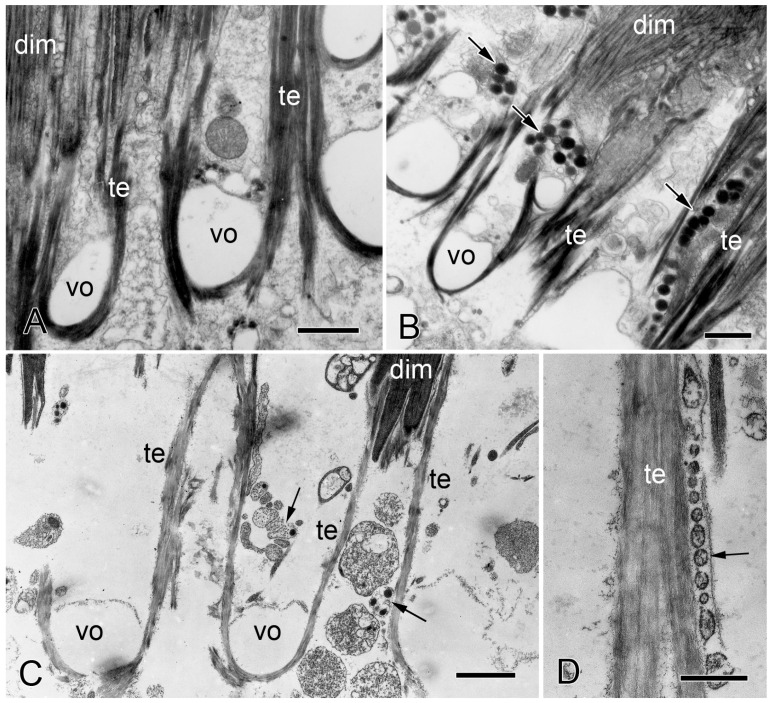
Transmission electron microscopy of the IMTs of *O. nigra*. Horizontal sections through non-autotomy and autotomy attachment regions of dorsal intervertebral muscles (dim). te, IMT; vo, decalcified stereom. (**A**) Proximal non-autotomy attachment region showing the absence of juxtaligamental cell components. Scalebar = 1 µm. (**B**) Distal autotomy attachment region showing the presence of juxtaligamental cell components (arrows). Scalebar = 1 µm. (**C**,**D**) Autotomizing attachment region. (**C**) Note the elongated tendon fibers and juxtaligamental cell components (arrows). Scalebar = 1.5 µm. (**D**) Row of vesicles (arrow) in close contact with elongated tendon fiber. Scalebar = 0.5 µm. From reference [4]. Used with permission of John Wiley & Sons—Books, conveyed through Copyright Clearance Center, Inc.

**Figure 9 biology-13-00375-f009:**
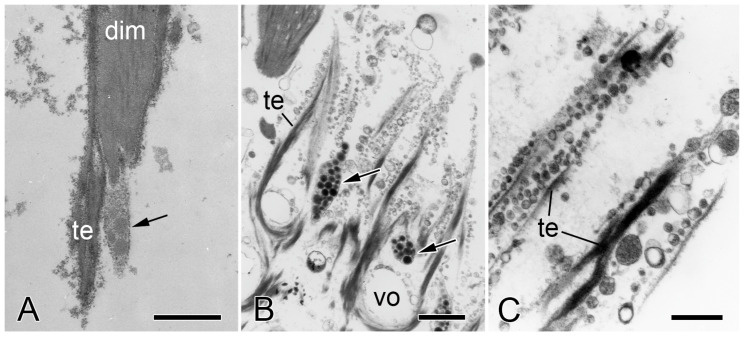
Transmission electron microscopy of the IMTs of *O. nigra*. Horizontal sections through the distal autotomy attachment region of dorsal intervertebral muscles after autotomy. (**A**) Tip of a detached muscle cell (dim) to which fragments of IMT (te) and JLCP (arrow) are attached. vo, decalcified stereom. Scalebar = 1 µm. (**B**) Ruptured IMTs and juxtaligamental components (arrows). Scalebar = 1 µm. (**C**) Vesicles in close contact with ruptured IMTs. Scalebar = 0.5 µm. (**A**) From reference [57]. Used with permission of John Wiley & Sons—Books, conveyed through Copyright Clearance Center, Inc.

**Figure 10 biology-13-00375-f010:**
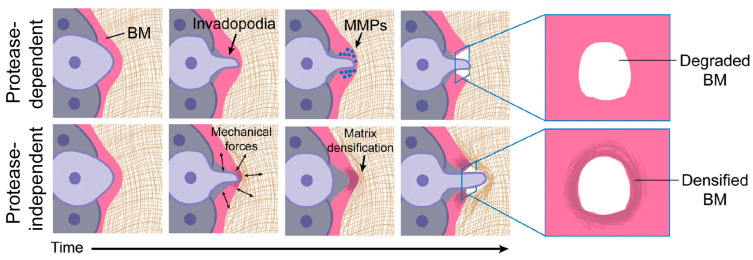
Involvement of invadopodia in protease-dependent and protease-independent BM invasion. In protease-dependent invasion, the BM is breached only by the enzymatic action of MMPs. In protease-independent invasion, the BM is breached by mechanical forces generated during repeated cycles of invadopodial protraction and retraction, with compressive forces displacing and densifying BM components adjacent to the breach. These processes occur over several hours. Adapted from reference [11] https://doi.org/10.1083/jcb.201903066 (accessed on 13 February 2024), under the terms of the Creative Commons License, https://creativecommons.org/licenses/by-nc-sa/4.0/, (accessed on 13 February 2024).

**Table 1 biology-13-00375-t001:** Hierarchy of organizational features of supramolecular structures, adapted from [12] and applied to BMs. TEM, transmission electron microscope.

Feature	Supramolecular Structure	Basement Membranes
Composition	Numbers and kinds of molecules	Collagens IV, XV and XVIII, laminin, nidogen, perlecan, agrin etc.
Constitution	Connections between the molecules	Intermolecular stabilization by covalent (collagen IV network only) and non-covalent bonds
Configuration	Orientation of neighboring molecules about each other in 3D space	Independent collagen IV and laminin networks linked to each other and cell surface by nidogen, perlecan, agrin etc.
Conformation	Overall shape of supramolecular structure in 3D space	Observed in TEM as lamina densa (comprising finely granular and filamentous material) and lamina lucida

**Table 2 biology-13-00375-t002:** Histochemistry of IMTs and intervertebral ligament of the brittlestar *Ophiocomina nigra*. CSA, chondroitin sulfate A; CSC, chondroitin sulfate C; HA, hyaluronic acid; β, β-metachromasia; γ, γ-metachromasia; −, negative reaction; +, weak positive reaction; ++, moderate positive reaction; +++, strong positive reaction [4,56].

Test	Detects	Reaction	
		Tendon	Ligament
PAS after amylase	neutral 1,2 glycols	+++	++
Toluidine blue (alcoholic)	acidic groups	β	β
Toluidine blue (aqueous)	acidic groups	γ	γ
Toluidine blue after hyaluronidase	CSA, CSC, HA	−	−
Alcian blue, pH 1.0	sulfate groups	+	++
Alcian blue, pH 2.5	carboxyl groups	+ ^1^	+

^1^ This was wrongly reported as ++ in reference [4].

**Table 3 biology-13-00375-t003:** Stiffness (Young’s modulus) of (A) BMs and (B) other collagenous structures. Most values in (A) were obtained by nano-indentation and atomic force microscopy; the exceptions were obtained by ^1^ inflation techniques that imposed uniaxial strain, ^2^ an inflation technique that imposed biaxial strain, and ^3^ direct tensile loading that imposed uniaxial strain. CDL, compass depressor ligament; ILM, inner limiting membrane of eye; MCV, mesenteric capillaries and venules.

**(A) Basement Membranes**			
**Animal**	**BM Location**	**Stiffness MPa**	**Reference**
Cat ^1^	MCV	1.8–5.4	[85]
Cat ^2^	Lens capsule	0.82–7.74	[86]
Chick	ILM	0.95–3.30	[79]
Human	ILM	1.5–5	[45]
Human	ILM	0.024	[87]
Human	Lens capsule	3.92–4.37	[45]
Human	Anterior cornea	0.002–0.015	[88]
Human	Descemet’s membrane	0.02–0.08	[88]
Mouse (neonatal)	ILM	3.81	[79]
Mouse (adult)	ILM	4.07	[79]
Mouse ^3^	Renal tubule	0.438–3.230	[89]
Mouse	Mesentery	0.055	[90]
Rabbit	Anterior cornea	0.0045	[84]
Rabbit	Descemet’s membrane	0.0117	[84]
Rabbit ^1^	Renal tubule	7–10	[91]
*Drosophila*	Egg chamber	0.03–0.07	[92]
*Drosophila*	Egg chamber	0.02–0.8	[93]
*Drosophila* ^3^	Malpighian tubule	1.4	[94]
**(B) Other structures**			
**Animal**	**Structure**	**Stiffness MPa**	**Reference**
Cow	Tendon (extra-ocular)	59	[95]
Dolphin	Tendon (sacrocaudalis)	1430	[96]
Human	Tendon (tibialis anterior)	450–1200	[97]
Human	Adventitia (arterial)	1.30–1.43	[98]
Human	Corneal stroma	0.033	[99]
Human	Dermis	0.03–0.15	[100]
Human	Dermis	0.1–18.4	[101]
Rabbit	Corneal stroma	0.0004–0.0095	[84]
Sea cucumber	Dermis	0.3–3	[102]
Sea urchin	CDL	1.43	[70]
Tuna	Tendon (caudal)	1310	[103]
Rat	Collagen fibril	39–130	[104]
Sea cucumber	Collagen fibril	360–1600	[105]
Cow	Collagen I molecule	2900	[106]
Rat	Collagen I molecule	5100–9000	[106]

**Table 4 biology-13-00375-t004:** Ultimate tensile strength (UTS) of (A) BMs and (B) other collagenous structures. CDL, compass depressor ligament; IAL, intervertebral arm ligament.

**(A) Basement Membranes**			
**Animal**	**BM Location**	**UTS MPa**	**Reference**
Cat	Lens capsule	0.17	[86]
Human	Anterior lens capsule	1.5–17.5	[82]
Rabbit	Renal tubule	1.8–2.0	[91]
Rabbit	Alveolar capillary	0.8	[107]
**(B) Other structures**			
**Animal**	**Structure**	**UTS MPa**	**Reference**
Dolphin	Tendon (sacrocaudalis)	62–95	[96]
Human	Tendon (calcaneal)	60	[108]
Human	Adventitia (arterial)	1.30–1.43	[98]
Brittlestar	IAL	6.17	[109]
Sea urchin	CDL	0.14	[70]
Tuna	Tendon (caudal)	22–33	[103]
Rat	Collagen fibril	39–130	[104]
Sea cucumber	Collagen fibril	70–470	[105]
Atomistic model	Collagen molecule	11,200	[110]

## Data Availability

Not applicable.

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
