# Peer review of "Basement Membranes, Brittlestar Tendons, and Their Mechanical Adaptability"

_biology, 2024, doi:10.3390/biology13060375_

Round 1

Reviewer 1 Report

Comments and Suggestions for Authors

Dear Authors,

The manuscript is very interesting, but please compare, contrast, and discuss more invertebrates to compare with echinoderms, especially other classes than ophiuroids. 

Please revise the manuscript accordingly. 

Best regard,

Reviewer

Author Response

RESPONSE TO REVIEWER 1

I thank Reviewer 1 for their comments.

Point 1.

The manuscript is very interesting, but please compare, contrast, and discuss more invertebrates to compare with echinoderms, especially other classes than ophiuroids.

Response.

The BMs of only two invertebrates (Caenorhabditis elegans and Drosophila melanogaster) have been investigated in detail. These are already mentioned frequently in the text (on pages 3-5, 15-17). I have, however, now added a mention of placozoan and cnidarian BMs on page 3. Regarding echinoderms other than ophiuroids, information on the BM organization of the other echinoderm classes is already provided on page 9.

Please note that all the following points refer to comments in the pdf version provided by Reviewer 1.

Point 2.

Page 2.

The non-echinoderm BMs description could be better same invertabrate not vertebrate.

Response.

There is much more information available on vertebrate BMs than on invertebrates BM and only two invertebrate BMs have been investigated in detail (see above). Also, echinoderms and chordates (including vertebrates) are sister groups in the Superphylum Deuterostomia. Echinoderms are evolutionarily closer to vertebrates than to other invertebrates and therefore it is more relevant to compare echinoderm BMs with vertebrate BMs than with those of other invertebrates.

Point 3.

Page 4.

how's laminins in invertebrate? it could be very different.

Response.

Echinoderm laminins are already discussed on page 9.

Point 4.

Page 4.

and how's about it in other invertebrates?

Response.

I am not aware of relevant information on other invertebrate laminins.

Point 5.

Page 5.

How's perlecan in other invertebrates?

Response.

Echinoderm perlecans are already mentioned on page 5.

Point 6.

Page 6.

of human epidermis?

Response.

I have clarified this by adding “of animal organs”.

Point 7.

Page 7. Figure 4.

what do red arrows mean?

Response.

This is already explained in lines 9 and 10 of the caption.

Point 8.

Page 8. Figure 5.

what's the big arrow?

what’s m?

Response.

As already explained in line 4 of the caption, the big (“fat”) arrows indicate the plane of autotomy.

“m” refers to the dorsal intervertebral muscles, which are wrongly labelled as “dim” in the caption. This has been corrected.

Point 9.

Page 8.

what treatment? please describe.

Response.

The treatment is “with Clostridium histolyticum type III collagenase”, as already described in the text.

Point 10.

Page 9.

it would be better to compare with other invertebrate.

Response.

Please see the response to Point 2 above.

Point 11.

Page 13.

vo?

Response.

An explanation has been added to the caption.

Point 12.

Page 14.

Please describe BM in invertebrate more than in vertebrate.

Response.

There is relatively little information available on the biomechanics of invertebrate BMs, and most of that refers to C. elegans and D. melanogaster, discussion of which is already included in the text in Sections 3.1.1 (including Table 2A) and 3.1.2. Please also see the response to Point 2 above.

Point 13.

Page 16.

invertebrate is more related rather than human/vertebrate

Response.

Please see the response to Point 2 above and note that C. elegans and D. melanogaster BMs are already discussed in this Section.

Point 14.

Page 21.

Need more brief conclusion, no more reference citation

Response.

I have tried to make the ‘Conclusions’ as brief as possible, but need to cover a number of important points, none of which I would like to omit. Please note that I have introduced only three additional references in this Section.

Reviewer 2 Report

Comments and Suggestions for Authors

This manuscript provides a comprehensive overview of basement membranes in non-echinoderm animals and of brittlestar tendons. The content is articulated with clarity and precision, making for an engaging read.

1. Figure 9, the author can consider adding a time scale in this figure.

2. The author can consider expanding on how BMs and IMTs could lead to novel therapeutic approaches, and consider discussing any limitations or challenges in this area of study.

Author Response

RESPONSE TO REVIEWER 2

I thank reviewer 2 for their comments.

Point 1.

Figure 9, the author can consider adding a time scale in this figure.

Response.

It is not possible to add a scale as two different processes are depicted in the Figure. However, to give an indication of the timescale, I have added to the caption: “These processes occur over several hours”.

Point 2.

The author can consider expanding on how BMs and IMTs could lead to novel therapeutic approaches, and consider discussing any limitations or challenges in this area of study.

Response.

Because “novel therapeutic approaches” refers to possible and as yet unknown mechanisms of BM destabilization, it is not possible to provide more detail here. In addition, this would be incompatible with Reviewer 1’s suggestion that the ‘Conclusions’ should be more brief.